The role of andrographolide as a potential anticancer agent against gastric cancer cell lines: a systematic review

Anis Amiera Muhamad Alojid 1
Tuan Kub Tuan Noorkorina 1 tnkorina@gmail.com
Harun Azian 1
Mohamud Rohimah 2
Razian Nur Rina Alissa 1
http://orcid.org/0000-0002-2059-821X Muhammad Ismadi Yasmin Khairani 1
Wan Alias Wan Alif Syazwani 1
Abdulrazak Mohammed Hussain 1
Teni Ernawati 3
1 Department of Medical Microbiology & Parasitology, School of Medical Sciences, Universiti Sains Malaysia , Kota Bharu, Kelantan , Malaysia
2 Department of Immunology, School of Medical Sciences, Universiti Sains Malaysia , Kota Bharu, Kelantan , Malaysia
3 National Research and Innovation Agency (BRIN), Research Centre for Pharmaceutical Ingredients and Traditional Medicines , Yogjakarta , Indonesia
Wang Liang
Electronic publication date: 2024 Nov 20
Publication date: 2024
Volume: 12
Electronic Location ID: e18513
Received 2024 Jun 25; Accepted 2024 Oct 21
Copyright: © 2024 Anis Amiera et al.
Copyright year: 2024
Copyright holder: Anis Amiera et al.
License: This is an open access article distributed under the terms of the Creative Commons Attribution License, which permits unrestricted use, distribution, reproduction and adaptation in any medium and for any purpose provided that it is properly attributed. For attribution, the original author(s), title, publication source (PeerJ) and either DOI or URL of the article must be cited.
License URL: https://creativecommons.org/licenses/by/4.0/

Keywords: Andrographis paniculata, Andrographolide, In vitro, Gastric cancer cell lines, Anticancer

Funding: Malaysian Ministry of Higher Education FRGS/1/2022/SKK12/USM/03/5 This work was supported by the Malaysian Ministry of Higher Education for the financial support via a Fundamental Research Grant Scheme (FRGS/1/2022/SKK12/USM/03/5) for this study. The funders had no role in study design, data collection and analysis, decision to publish, or preparation of the manuscript.

==============================
Objective

To critically analyse literature on the anticancer properties of andrographolide in in vitro studies on gastric cancer cells.

Method

This study systematically reviewed articles from 2013 to 2024 across five prominent databases; PubMed, Google Scholar, Web of Science, Scopus, and Science Direct, EMBASE, Cochrane library and DOAJ. The study eligibility criteria include original studies assessing using gastric cancer cell lines and articles utilizing extracted andrographolide from Andrographis paniculata or standard andrographolide source treatment. The following exclusion criteria were articles written in a different language, review articles, book chapters, conference articles, scientific reports. Duplicated articles were removed using Mendeley software.

Result

Out of 93 articles, six were relevant, primarily focusing on in vitro analyses with gastric adenocarcinoma cell lines.

Conclusion

These studies indicate that andrographolide can hinder the cell cycle, suppress cell proliferation, alleviate oxidative stress, and induce apoptosis by prompting gastric cancer cells to undergo self-destruction, which is a crucial mechanism for controlling and eliminating cancerous growths.

Introduction

Cancer poses a significant global public health challenge. According to the International Agency for Research on Cancer, the number of cancer patients worldwide is projected to reach 28.4 million by 2040, a 47% increase from 2020, solidifying its position as one of the leading causes of death (Hu et al., 2024). Among all cancer types, gastric cancer ranks fifth in global diagnoses and is the third leading cause of cancer-related deaths, accounting for over a million new cases annually. It carries particularly high mortality rates in regions such as South America, Eastern Europe, and East Asia (Smyth et al., 2020; Bray et al., 2018). Gastric cancer, commonly known as stomach cancer, arises from the stomach lining and often progresses without early symptoms, making early detection challenging. Key risk factors include Helicobacter pylori infection, dietary choices, smoking, and genetic susceptibility (Mamun, Younus & Rahman, 2024). The prognosis for gastric cancer remains poor, especially in cases diagnosed at later stages. With the high hospital burden, severe oncological and surgical resource constraints, chemotherapy treatment is only applied to advanced gastric cancer cases with poor postoperative therapy compliance. In certain cases, adjuvant chemotherapy is prioritised to patients with a high risk of recurrence. Furthermore, the existing recommended chemotherapy treatments are not always successful due to the adverse effects of chemotherapeutic agents. The lack of effective anticancer with immunotherapeutic and antibiotic agents contributes to the treatment failure.

In recent years, there has been growing interest in exploring natural compounds as potential anticancer agents. Andrographolide is a novel immunotherapeutic agent extracted from Andrographis paniculata (Burm. F.) Nees (family Acanthaceae) and is widely used as a traditional medicine in Asian countries. Andrographolide gets attention for the treatment of gastric cancer due to its anti-inflammatory, antibacterial, immunomodulatory, and anticancer role in various cancers including in melanoma, leukaemia, glioblastoma, breast, lung, oesophageal, colorectal, bladder, pancreatic, and liver cancer (Yadav, Sadhukhan & Saha, 2022). Furthermore, previous studies have reported the potential role of andrographolide in the regulation of oxidative stress, apoptosis, necrosis, autophagy, inhibition of cell adhesion, proliferation, migration, invasion, and angiogenesis (Yadav, Sadhukhan & Saha, 2022). Various anti-gastric cancer-related signalling pathways, such as Src/MAPKs/AP-1 (Yuan et al., 2018) and TLR4/NF-kB/MMP-9 (Zhang et al., 2017) are also controlled by andrographolide. A study by Gao et al. (2021) reported that andrographolide induced p53-mediated apoptosis in SGC7901 gastric carcinoma cells which adds to a novel approach in anticancer therapies. To date, no clinical trials have been conducted specifically on the use of andrographolide for treating gastric cancer. However, one clinical trial explored the effectiveness and protective effects of andrographolides combined with capecitabine in elderly patients with locally advanced, recurrent, or inoperable metastatic colorectal cancer. Unfortunately, this trial was terminated after phase 2 due to low accuracy in the results (Tundis et al., 2023). This review exploring the potential of different sources of andrographolide consistently exhibit anti-gastric cancer properties across various gastric cancer cell lines.

The literature exploring andrographolide potential in cancer treatment is extensive, but there is no systematic synthesis of in vitro studies elucidating the role of andrographolide as an anti-gastric cancer agent. This systematic review will offer valuable insights for future research, particularly since no in vivo or clinical trials involving andrographolide as a treatment for gastric cancer have been conducted to date. By systematically synthesizing the available in vitro evidence, this review aims to meticulously analyse the literature on the therapeutic characteristics of andrographolide in in vitro studies involving gastric cancer cells. The objectives encompass assessing evidence for anticancer effects towards in vitro gastric cancer cell lines, understanding underlying mechanisms, and summarizing research findings. These findings contribute to a deeper understanding of natural compounds like andrographolide, guiding future studies towards preclinical in vivo models and clinical trials to further investigate the therapeutic potential of andrographolide since in vitro studies do not replicate the complex tumour microenvironment that exists in vivo. Furthermore, it is crucial to interpret the result of studies utilizing various source of gastric cancer cell lines as findings may differ due to heterogeneity in cell lines characteristics.

Materials and Methods

Study design

The Preferred Reporting Items for Systematic Reviews and Meta-Analyses (PRISMA) criteria were used in the conduct of this systematic review (Page et al., 2021). The search procedure was completed by utilizing the database assessed on 10 September 2024; PubMed (https://pubmed.ncbi.nlm.nih.gov), Scopus (https://www.scopus.com). Science Direct (https://www.sciencedirect.com), Google Scholar (https://scholar.google.com), Web of Science (https://mjl.clarivate.com), DOAJ (https://doaj.org), Cochrane library (https://www.cochranelibrary.com) and EMBASE (https://www.elsevier.com).

This analysis included only research articles published in the English language. The search keywords that were used to access all databases were “Andrographis paniculata” OR “andrographolide” AND “gastric cancer” OR “gastric carcinoma” OR “stomach cancer” OR “stomach carcinoma”. Two authors (A.A.M.A and W.A.S.W.A) performed the search and selection of studies while a third reviewer (T.N.T.K) evaluated the sources in cases of disagreement.

Inclusion and exclusion criteria

The criteria were included in the selection process: original, full-text articles; research studies assessing using gastric cancer cell lines; and articles utilizing extracted andrographolide from AP or standard andrographolide source treatment. The following criteria were exclusion criteria: articles written in a different language; review articles, book chapters, conference articles or scientific reports; news, editorial letters, or social media pieces; and duplicated articles. The inclusion and exclusion criteria were categorized according to the population, intervention, comparison and outcome (PICO) as tabulated in Table 1.

Table 1 PICO element for inclusion and exclusion criteria.

PICO element	Inclusion	Exclusion	
Population	Studies of gastric cancer cell lines.	Studies that are not related to gastric cancer cell lines.	
Intervention	Studies that utilizing andrographolide as treatment.	Studies that are not utilizing andrographolide as treatment.	
Comparison	Not explicitly mentioned	Not explicitly mentioned	
Outcome	Studies that have recorded anticancer activity.	Studies that do not record anticancer activity.	

Quality assessment and risk of bias analysis

The QUIN tool is utilized for the quality assessment of the selected in vitro studies and it provides a formula to quantify the risk of bias (Sheth et al., 2022). Evaluations were based on 12 criteria as listed in Table 2. Studies are categorized as low risk of bias if scored >70%, medium risk of bias is between 50% to 70% while studies scored <50% are considered as high risk of bias. The critical judgement was made by two authors (A.A.M.A and W.A.S.W.A) and a third author (T.N.T.K) evaluated the data in case of disagreement.

Table 2 Scoring sheet for the QUIN tool (quality assessment tool for in vitro studies).

Adequately specified (score = 2), Inadequately specified (score = 1), Not specified (score = 0), Not applicable.

No	Criteria	Malat et al. (2021)	Dai, Wang & Pan (2017)	Monger et al. (2017)	Lim et al. (2017)	Ma et al. (2021)	Li et al. (2013)	
1	Clearly stated aims/objectives	2	2	2	2	2	2	
2	Detailed explanation of sample size calculation	Not applicable	Not applicable	Not applicable	Not applicable	Not applicable	Not applicable	
3	Detailed explanation of sampling technique	Not applicable	Not applicable	Not applicable	Not applicable	Not applicable	Not applicable	
4	Details of comparison group	2	2	2	2	2	2	
5	Detailed explanation of methodology	2	2	2	2	1	2	
6	Operator details	2	2	2	2	2	2	
7	Randomization	2	2	2	2	2	2	
8	Method of measurement of outcome	2	2	2	2	2	2	
9	Outcome assessor details	2	2	2	2	2	2	
10	Blinding	Not applicable	Not applicable	Not applicable	Not applicable	Not applicable	Not applicable	
11	Statistical analysis	2	2	2	2	2	2	
12	Presentation of results	2	1	2	1	2	2	

Data extraction and analysis

The screening process and figure creation details are outlined in Fig. 1. Biorender and Microsoft PowerPoint were used for figures and tables to avoid copyright issues.

Figure 1 PRISMA flow diagram of literature selection process used in this review, applied to PubMed, Scopus, Science Direct, Google Scholar, Web of Science, EMBASE, Cochrane library and DOAJ databases.

Results

Selected articles and characteristics of studies

The search identified 93 relevant articles from five databases, Science Direct (39), followed by Scopus (22), PubMed (15), Google Scholar (10), and Web of Science (seven). Only six articles met the inclusion criteria after applying selection and exclusion criteria (Fig. 1).

The selected publications articles were issued in indexed journals, along with their corresponding journal impact factors (JIF) as per InCites Journal Citation Reports effective January 15, 2024: Biomed Research International (JIF 3.246), Asian Pacific Journal Cancer Prevention (JIF 2.514), Oncology Letters (JIF 2.9), Chinese Medical Journal (JIF 6.1), and Anticancer Research Journal (JIF 2.0), while MDPI Microorganism did not have a listed JIF.

The studies summarized in this review, inclusive of their general characteristics, are outlined in Table 3. Several research groups analysed andrographolide samples acquired from different suppliers utilizing diverse preparation techniques. For example, two studies employed andrographolide from Sigma, Burlington, MA, USA (Dai, Wang & Pan, 2017; Lim et al., 2017) whereas others explored combinations with different compounds utilizing sodium butyrate (NaB) (Malat et al., 2021) or recombinant human tumour necrosis factor-related apoptosis-inducing ligand (TRAIL) (Lim et al., 2017). One study compared parent andrographolide with its analogue where 0.1 to 50 µM andrographolide was extracted from the dried aerial part of Andrographis paniculata, alongside 19-triisopropyl-andrographolide analogue for comparison (Monger et al., 2017). In a separate study, EP80™ Andrographis paniculata extract standardized to 80% andrographolide content was dissolved in dimethyl sulfoxide (DMSO). Andrographis EP80™ is a natural product that dispenses 80 mg of andrographolide (Ma et al., 2021). Noteworthily, two studies did not specify the andrographolide source (Li et al., 2013; Malat et al., 2021).

Table 3 Reports of in vitro effects of andrographolide on gastric cancer cells.

Treatment	Experimental model	Protocol	Result obtained	Reference	
Andrographolide,
NaB, combination of andrographolide and NaB
(the source of andrographolide not mentioned)	AGS and AGS-EBV (kindly provided by Prof. Hironori Yoshiyama, Shimane University, Japan)	Cell cytotoxicity measured through CCK-8 assay.

Protein extraction analysis, treated with 25% andrographolide and protein concentration was determined using Bio-rad protein assay-dye reagent concentration kit.

Proteomic analysis by mass spectrometry.

Molecular docking was performed to confirm protein-protein interaction.

Quantification of gene expression by real time (RT)-PCR

Cell apoptosis was detected by flow cytometry.

	Cell viability is reduced when exposed to andrographolide and combination of andrographolide and NaB in comparison to AGS cell line as opposed to those cells subjected to NaB treatment only.

BALF4, BALF5, BMRF1, BBLF3, BRLF1, and BZLF1 suppressed.

LMP2A inhibited, EBNA1 and LMP1 unaffected.

Apoptosis was observed in both AGS and AGS-EBVcell lines.

	Malat et al. (2021)	
10 mg/ml of andrographolide (Sigma, Burlington, MA, USA)	SGC7901 cell line (purchased from Shanghai cell bank of the Chinese Academy of Sciences)	In vitro cytotoxic activity was assessed by MTT assay.

Cell cycle and apoptosis analysis was detected by flow cytometry.

Cell migration activity was examined by wound healing assay.

Number of transferred cells was analysed by transwell assay.

Proteins associated with invasion and apoptosis was performed by western blot.

Protease activity of MMP-9 and MMP-2 was detected by gelatin zymography analysis.

Gene expression was measured by RT-PCR.

	5, 20, and 40 µg/ml andrographolide exhibited a significant reduction in the proportion of proliferative cells when compared to the control DMSO (p < 0.01).

40 and 20 µg/ml andrographolide caused cycle arrest at G2/M phase at 36.2% and 24.39% respectively.

Apoptosis ratio were 28.4%, 19.9%, and 16.5% respectively.

Cell migration activity decrease.

Cell invasion decreased in a dose-dependent manner.

MMP-2 and MMP-9 activities reduced.

TIMP-1/2, Bax, and Bik upregulated whereas MMP-2, MMP-9, CD147, Bcl-2 were downregulated.

Cyclin B1, phosphorylated CDC2, TIMP-1, TIMP-2, CD147, Bcl-2 and Bax were modulated.

	Dai, Wang & Pan (2017)	
0.1–50 µM andrographolide was isolated from the dried aerial part of Andrographis paniculata and 19-triisopropyl-andrographolide analogue (analogue-6).	Human GC cell line MKN-45 and AGS (purchased from Japanese Collection of Research Bioresources Cell Bank and ATCC)	In vitro cytotoxic activity was assessed by MTT assay

Apoptosis was measured by flow cytometry.

Expression of PARP-1, cleaved PARP-1, Caspase 3, and p53 were analysed by western blot assay.

	IC50 value at 48 h was 6.3 ± 0.7 µM in MKN-45 cells and 1.7 ± 0.05 µM in AGS cells, Parent andrographolide ≥ 50 µM in MKN-45 cells and 11.3 ± 2.9 µM in AGS cells respectively.

Gamma-H2AX increased in MKN-45 cells with a concentration of 10 µM analogue (p < 0.01), while parent andrographolide did not induce DNA damage. Topo Ila enzyme significantly decreased in AGS cells with a concentration of 2 µM analogue (p < 0.01).

Analogue-6 more effective in inducing apoptosis compared to the parent andrographolide. At 48 h analogue concentrations 1, 2, & 5 µM induced apoptosis of AGS cells up to 10.7 ± 0.5, 28 ± 4.6 and 60.4 ± 6.5% respectively.

Significant increase (p < 0.01) in the expression of cleaved PARP-1 protein resulting from DNA damage and an increase in caspase 3 activation, which is indicative of apoptosis (p < 0.01). No change in p53 expression.

	Monger et al. (2017)	
10–50 µM andrographolide (Sigma-Aldrich; Merck KGaA, Darmstadt, Germany) alone or with 5–20 ng/ml recombinant human TRAIL (rhTRAIL; a gift from T.H. Kim, Department of Biochemistry and Molecular Biology, Chosun University, Korea)	Human GC cell lines SNU601, SNU638 and AGS were obtained from the Korean Cell Line Bank (Seoul, Korea)	In vitro cytotoxic activity was assessed by MTT assay.

Apoptosis was detected by Hoechst 33,342 (HO)/propidium iodide (PI) double staining.

Immunoblotting was done to measure the expression levels of cell cycle inhibitory protein and apoptosis-inducing proteins.

Cell proliferating activity was assessed by clonogenic assay.

Detection of reactive oxygen species (ROS) generation

	Cell growth and cell viability decreased.

Maximum apoptotic death occurred when 20 µM of andrographolide was present, while andrographolide triggered non-apoptotic cell death when the concentration exceeded 20 µM.

Expression of cyclin inhibitor p21, p27 protein in SNU638 cells, membrane death receptors DR4 (10–20 µm), and DR5 (30–40 µm) were induced. Andrographolide increased p53 levels in AGS and SNU638 cells was increased, but not in SNU601 cells.

Clonogenic activity was significantly reduced when a combination of rhTRAIL and andrographolide.

ROS increased.

	Lim et al. (2017)	
Andrographis (EP80™ Andrographis extract standardized to 80% andrographolide content, dissolved in DMSO) was purified by EuroPharma-USA and kindly provided by Professor Ajay Goel at the City of Hope Comprehensive Cancer Center.	GC cell lines MKN74 and NUGC4 were provided by the Cell Resource Center of Biomedical Research, Institute of Development, Aging and Cancer (Tohoku University, Sendai, Japan)	Cell viability and cell proliferation were measured by WST assay

Cell colony formation activity was measured through cell colony formation assays.

Apoptosis was measured by flow cytometry.

mRNA expression was analysed by RT-PCR.

HMOX1, GCLC, GCLM, and b-actin were analysed by western blot.

	Cell proliferation was suppressed in a dose-dependent manner at concentrations of 10, 20, 40, 60, 80, 100 µg/ml.

Significant inhibition of growth in both cell lines at concentration of 40 µg/ml andrographolide (p < 0.0001).

Size and number of colonies reduced compared to control.

Percentage of apoptotic cells significantly increased to 5.45 ± 0.95% for MKN74 cells and 6.18 ± 1.6% for NUGC4 cells.

Percentage of live cells significantly reduced to 19.43 ± 1.13% for MKN74 cells and 15.42 ± 0.65% for NUGC4 cells.

At the mRNA level, andrographolide significantly upregulated all target genes (p < 0.0001).

At the protein level, andrographolide increased the expression of HMOX-1 (p < 0.05), GCLC (p < 0.05) and GCLM (p < 0.05).

	Ma et al. (2021)	
Andrographolide
(the source of andrographolide not mentioned)	Human BGC-823 gastric epithelial cell line was obtained from Shanghai Jiao Tong University (Shanghai, China)	In vitro cytotoxic activity was measured by MTT assay

Cell cycle analysis to determine the number of cells at each cell cycle and the rate of apoptosis were measured by flow cytometer.

Assessment of apoptosis using Annexin-V/PI double-staining assay.

Cell morphology was observed using transmission electron microscopy.

Gene expression was analysed by RT-PCR.

Expression of Bcl-2, Bax, caspase-3 and B-actin were analysed by immunochemistry analysis.

	IC50 value is 35.3, 25.5, and 18 µg/ml for 24, 48, and 72 h respectively.

The percentage of cells in the G0-G1 phases increased, S and G2/M phase decreased.

The rates of early apoptosis in cells treated with 7.5, 10, and 15 µg/ml of andrographolide were 19.3%, 29.4%, and 52.7% respectively while the rate of late apoptosis were 10.8%, 10.9%, and 14.7% respectively. The apoptotic rate of andrographolide-treated cells was significantly higher than control (p < 0.05 Morphological changes including the condensation of chromatin, disintegration of nucleolus cytoplasmic vacuoles and necrosis after treatment with 10 ug/ml andrographolide Expression of Bax, and Caspase-3 proteins increased, while Bcl-2 decreased.

	Li et al. (2013)	

This review focused solely on the effect of andrographolide compound activity towards in vitro anticancer on human gastric cell lines. Human gastric adenocarcinoma (AGS) cells were used in two out of six studies, while others employed various human gastric cell lines such as SGC7901 (established from a human gastric adenocarcinoma patient), MKN-45 (derived from a gastric cancer patient in Japan), SNU601 (derived from the ascitic fluid of a Korean patient with gastric adenocarcinoma), SNU638 (established from a poorly differentiated gastric adenocarcinoma patient), MKN-74 (derived from a poorly differentiated gastric adenocarcinoma; surgical specimen), NUGC4 (established from a gastric cancer patient in Japan), and BGC-823 (derived from a gastric cancer patient in China) (Fig. 2). Concerning the evaluated parameters, the in vitro studies employed cytotoxic assays such as the MTT test where Dai, Wang & Pan (2017) assessed cytotoxicity on SGC7901 cells with concentrations from 5 to 40 µg/ml while Lim et al. (2017) studied anticancer activity on SNU601, SNU638, and AGS cell lines at 10 to 50 µM, Malat et al. (2021) examined the cytotoxicity activity of AGS and Eipstei-Barr Virus (EBV) cells using the CCK-8 test, and WST test studied by Ma et al. (2021) on MKN74 and NUGC4 cells, evaluating viability with concentrations from 10 to 100 µg/ml to ascertain the anti-gastric cancer efficacy of andrographolide.

Figure 2 Schematic representation of the principal gastric cancer cell lines tested to determine the anticancer effects of andrographolide.

Flow cytometry analysis was employed to evaluate cell cycle arrest and apoptosis (Li et al., 2013; Dai, Wang & Pan, 2017; Malat et al., 2021; Ma et al., 2021). Further, real-time PCR was employed for gene expression analysis (Li et al., 2013; Dai, Wang & Pan, 2017; Malat et al., 2021; Ma et al., 2021), whereas Western blot assays were applied to assess alterations in gene expression associated with apoptosis (Dai, Wang & Pan, 2017; Monger et al., 2017; Lim et al., 2017; Ma et al., 2021). The staining method involving Hoechst 33342/propidium iodide staining (Lim et al., 2017), and Annexin-V/PI double staining was combined with transmission electron microscopy to further examine apoptosis and visualize cell morphology (Li et al., 2013). In addition, Dai, Wang & Pan (2017) investigated cell migration through wound healing and transwell assays, alongside gelatine zymography analysis to detect MMP-9 and MMP-2 protease activity. Ma et al. (2021) assessed cell colony formation activity via colony formation assays. Only one study was focused on reactive oxygen species (ROS) generation in gastric cancer cells (Lim et al., 2017).

Effect of andrographolide on gastric cancer cell lines

The studies highlighted in Table 3 demonstrate the therapeutic advantages of andrographolide in the treatment of gastric cancer. Monger et al. (2017) conducted a comparison of the cytotoxic effects of parent andrographolide with analogue-6 on MKN-45 and AGS cells at 24, 48, and 72 h. Analogue-6 showed lower IC50 values (6.37 ± 0.7 µM for MKN-45 and 1.7 ± 0.5 µM for AGS) compared to parent andrographolide (>50 µM for MKN-45 and 11.3 ± 2.9 µM for AGS) at 48 h, indicating analogue-6 having greater cytotoxicity effect. Furthermore, Li et al. (2013) identified IC50 values of 35.3, 25.5, and 18 µg/ml for BGC-823 cells treated with andrographolide for 24, 48, and 72 h, respectively. Notably, the concentration of andrographolide lower than 7.5 µg/ml exhibited minimal inhibitory effect, while the concentration of andrographolide ranging from 15–60 µg/ml displayed the most substantial inhibition; no inhibitory effect was observed at a concentration exceeding 60 µg/ml.

Interestingly, researchers also investigated the cytotoxic effects of andrographolide on AGS-EBV cells, administering three treatments: andrographolide alone, a combination of andrographolide with NaB, and NaB alone, to both AGS and AGS-EBV cells. Findings indicated lower cell viability with andrographolide alone and the combination compared to NaB alone in both cell lines. AGS-EBV cells exhibited higher cytotoxicity than AGS cells upon andrographolide treatment, signifying its superior efficacy in AGS-EBV cells. Andrographolide prompted apoptosis in both cell lines, with significantly greater cell death observed in AGS-EBV cells compared to AGS cells (Malat et al., 2021).

Lim et al. (2017) explored the anticancer properties of andrographolide using in vitro model of gastric cancer cell lines of AGS cells, SNU638, and SNU601, revealing that andrographolide curtailed the growth rate and viability of these cell lines. Apoptosis assays unveiled andrographolide induced apoptotic and non-apoptotic cell death, peaking at 20 µM concentration with nuclei alterations. This investigation also unveiled that the combined treatment of andrographolide with rhTRAIL notably increased the number of apoptotic bodies compared to solely rhTRAIL treatment. Furthermore, Li et al. (2013) reported significant elevation in apoptotic rates induced by andrographolide in contrast to the control group (p < 0.05). Morphological changes including chromatin condensation, nucleolar disintegration, cytoplasmic vacuolization, and necrosis were also documented.

Additionally, andrographolide exhibited an increase in apoptotic cells to 5.45 ± 0.95% for MKN74 and 6.18 ± 1.6% for NUGC4 cells, alongside a decrease in live cells to 19.43 ± 1.13% for MKN74 and 15.42 ± 0.65% for NUGC4 cells. The inhibitory effect of andrographolide on proliferation in MKN74 and NUGC4 cell lines was dose-dependent, notably inhibiting growth at 40 µg/ml (p < 0.0001), reducing colony size and number, thus indicating its anti-tumorigenic effect (Ma et al., 2021). Treatment with andrographolide also triggered apoptosis in both AGS and AGS-EBV cell lines, in contrast to the DMSO (control) and NaB treatment (Malat et al., 2021). In SGC7901 cells, the induction of apoptosis by andrographolide resulted in an apoptosis ratio of 28.4%, 19.9%, and 16.5% respectively (Dai, Wang & Pan, 2017). Notably, a study by Monger et al. (2017) revealed that analogue-6 was more effective in inducing apoptosis compared to the parent andrographolide with analogue concentrations 1, 2, & 5 µM induced apoptosis of AGS cells up to 10.7 ± 0.5, 28 ± 4.6 and 60.4 ± 6.5% respectively.

Additionally, andrographolide induced alterations in the cell cycle, leading to a decrease in the percentage of cells in the S and G2/M phases in BGC-823 cells, while increasing the percentage of cells in the G0-G1 phase as reported by Li et al. (2013). Andrographolide inhibited SGC7901 cell proliferation from 5 to 40 µg/ml, inducing G2/M phase cell cycle arrest at 40 and 20 µg/ml, with decreased GC cells at the G1 phase (36.2% and 24.39% respectively), effectively impeding the cell cycle (Dai, Wang & Pan, 2017).

Moreover, andrographolide upregulated the expression of cyclin inhibitor p21 and p27 protein in SNU638 cells. It induced the expression of the membrane death receptors DR4 (at 10–20 µM) and DR5 (at 30–40 µM) expression and increased p53 levels in AGS and SNU638 cells, though not in SNU601 cells (Lim et al., 2017). At a molecular level, andrographolide significantly upregulated target genes at the mRNA level (p < 0.0001) and enhanced HMOX-1 (p < 0.05), GCLC (p < 0.05), and GCLM (p < 0.05) expression at the protein level (Ma et al., 2021). The upregulation of TIMP-1/2, Bax, and Bik, coupled with down-regulating MMP-2, MMP-9, CD147, Bcl-2 resulted in the suppression of proliferation, promotion of apoptosis, and inhibition of invasion by influencing proteins like cyclin B1, phosphorylated CDC2, TIMP-1, TIMP-2, CD147, Bcl-2, and Bax (Dai, Wang & Pan, 2017). Additionally, andrographolide upregulated pro-apoptotic proteins including Bax and caspase-3 while downregulating the anti-apoptotic protein Bcl-2 (Li et al., 2013).

Research findings suggest that andrographolide has shown anticancer effects against the gastric cancer cell lines that were tested, due to its cytotoxic properties. Different sources of andrographolide resulted in varying rates of inhibition, with andrographolide derivatives showing lower IC50 values (indicating greater cytotoxicity) compared to the original compound. Moreover, the synergistic effect of andrographolide with other compounds enhanced the cytotoxic activity of gastric cancer cell lines. The expression of pro-apoptotic proteins such as p21, p27, DR4, DR5, and p53 signalling pathway to regulate the apoptosis, cell cycle, and proliferation of tumour cells, thus playing a role in the treatment of gastric cancer.

Discussion

Andrographis paniculata, recognized for its abundant andrographolide compound, has been traditionally used in Southeast Asian countries to treat inflammatory disorders. Recently, herbal medicines have gained recognition in cancer treatment. Andrographolide, in particular, has emerged as a potential anticancer agent. Nevertheless, its precise role and molecular mechanisms in human cancer treatment remain unclear. Gastric cancer, a common gastrointestinal malignancy, involves complex invasion and metastasis processes driven by multiple genes. Understanding these molecular mechanisms is crucial for developing effective anticancer therapies. Herbal drugs, including andrographolide, are gaining attention for their promising effects in this context.

In the articles reviewed, the efficacy of andrographolide was investigated in vitro using various types of gastric cancer cell lines, as presented in Table 3 and Fig. 1. These included cell lines derived from AGS, SNU601, SNU638, MKN74, NUGC4, SGC7901, and BGC-823. The cell lines models were employed in this study because there is no study has been done to investigate the effect of andrographolide as an anti-gastric treatment in animal models. Among these, AGS cell lines were predominantly employed in three out of six studies. AGS cells were originally derived from a gastric adenocarcinoma, a stomach cancer patient. Since adenocarcinoma is the most common type of gastric cancer, AGS cells provide a relevant model for studying this type of cancer. Other than that, AGS cells exhibit tumour-like behaviours that enable researchers to investigate the mechanisms of gastric cancer progression, metastasis, and treatment response (Tu, Jin & Shi, 2003). Andrographolide showed apoptotic, anti-proliferative, and cytotoxic activity against all tested gastric cell lines. This finding is significant as it demonstrates that andrographolide exhibits anticancer properties across all tested cell lines, highlighting its potential as a chemotherapeutic agent that can effectively target various aspects of gastric cancer. Studies indicate andrographolide’s pivotal role is inducing apoptosis. All six reviewed studies reported andrographolide’s cytotoxic effects in gastric cell lines by promoting apoptosis, causing cell cycle arrest, and activating ROS pathways (Malat et al., 2021; Dai, Wang & Pan, 2017; Monger et al., 2017; Lim et al., 2017; Ma et al., 2021; Li et al., 2013).

Regarding the apoptotic effects of andrographolide, research has demonstrated its ability to modulate several signalling pathways associated with gastric cancer. A study by Li et al. (2013) demonstrated the apoptotic effect in BGC-823 cells by upregulating Bax and caspase-3 expressions while downregulating Bcl-2. Similarly, Dai, Wang & Pan (2017) revealed that andrographolide treatment in SGC7901 cells upregulated Bax and several target genes including TIMP-1/2, cyclin B1, P-Cdc2, and Bik, while downregulating Bcl-2 and MMP-2/9 expressions. The potential of andrographolide lies in its ability to impede the progression of gastric cancer by suppressing MMP-2 and MMP-9 activities simultaneously enhancing TIMP-1 and TIMP-2 expression. This dual action is crucial for impeding cancer cell migration (Cabral-Pacheco et al., 2020).

Monger et al. (2017) examined the anticancer properties of 19-triisopropyl andrographolide (analogue-6), a derivative of andrographolide, against human gastric cancer cells. The analogue displayed robust anticancer activities by primarily targeting the DNA Topo IIα enzyme and inducing cellular apoptosis. Its mechanism of action involved inhibiting the Topo IIα enzyme, resulting in DNA impairment, PARP-1 cleavage, and enhanced caspase three activity, ultimately inducing apoptosis via a p53-independent mechanism. These discoveries highlight the potential of analogue-6 as a promising chemotherapeutic agent for gastric cancer.

The potential of andrographolide as an anticancer agent was further studied by Ma et al. (2021); they found that EP80™ andrographis extract, a natural product rich in andrographolide, upregulated ferroptosis-related genes like HMOX1, GCLC, and GCLM. Ferroptosis, a form of regulated cell death dependent on iron and ROS, presents a novel therapeutic avenue for gastric cancer (Li et al., 2019). rhTRAIL is a cytokine belonging to the tumour necrosis factor (TNF) family, and it is responsible for its role in inducing apoptosis in cancer cells with little effect on normal cells. Andrographolide upregulates the expression DR5, enhancing the binding sites available for rhTRAIL. This increased expression of death receptors improves the sensitivity of gastric cancer cells to TRAIL-induced apoptosis. In a study by Lim et al. (2017), the role of ROS in apoptosis induced by a combination of rhTRAIL was explored. Exposure of andrographolide substantially escalated ROS levels in AGS cells. Considering this evidence, it is justifiable to carry out more studies utilizing combinations of andrographolide and other compounds, since several studies reviewed could lead to synergistic effects and anticancer effects (Lim et al., 2017; Malat et al., 2021).

Furthermore, extensive research on AGS-EBV cells has indicated that andrographolide effectively triggers cytotoxicity and apoptosis in both AGS and AGS-EBV cell lines by elevating the expression of apoptosis-related protein. Moreover, andrographolide inhibits EBV lytic reactivation by targeting host transcription factors, partly through HDAC6 and TF interactions, leading to cell apoptosis. The upregulation of pro-apoptotic proteins like BCL2L1, ENDOG, and PUMA suggest mitochondria-dependent pathway involvement. These findings highlight andrographolide’s potential as an effective anticancer treatment, particularly for EBV-positive gastric cancer cells (Malat et al., 2021).

In the context of andrographolide-induced cell cycle arrest, Li et al. (2013) conducted a study on BGC-823 cells, where they noted a reduction in the cell percentage following treatment with andrographolide. Moreover, there was increasing G0-G1 phase cell arrest in a concentration-dependent manner. This direct anticancer mechanism of action entailed the induction cell cycle arrest specifically at the G0-G1 phase through the upregulation of p27 and the downregulation of cyclin-dependent kinase 4 (CDK4). Additionally, andrographolide was found to augment the production of tumour necrosis factor-alpha and the expression of CDK4 marker, boosting lymphocyte cytotoxicity against cancer cells indirectly. In line with these findings, Lim et al. (2017) found increased sub-G1 phase cells, indicating apoptosis in AGS cells, while Dai, Wang & Pan (2017) observed G2/M phase arrest in BGC-823 cells.

In our study, we found that andrographolide is a multitarget anticancer agent. We predicted that the mechanism of action of andrographolide in the treatment of gastric cancer is mediated by the regulation of pro-apoptotic proteins and other signalling pathways to regulate the proliferation, apoptosis, and migration of tumour cells, thus playing a role in the treatment of gastric cancer. Hence, it is promising to discover more novel anticancer properties of this compound. This study provides a rationale for using andrographolide for the treatment of gastric cancer.

Conclusions

This study extensively analysed existing data on andrographolide and its effects on gastric cancer, focusing specifically on in vitro studies. In the tested gastric cancer cell lines, andrographolide has demonstrated significant anticancer activity through various mechanisms of action. These include cytotoxic activity, induction of apoptosis, cell cycle arrest, and modulation of the immune response through the signalling pathways involved in the development of gastric cancer cells. Moreover, some studies have examined the combination of andrographolide with other compounds such as NaB and rhTRAIL onto gastric cancer cell lines. Different sources of andrographolide have shown diverse findings regarding its anti-gastric cancer activities. Further exploration using in vivo models is recommended to mimic the tumour microenvironment and study the interactions in cancer progression, metastasis and response to the treatments. Additional preclinical trials are crucial to authenticate the anticancer properties of andrographolide. This comprehensive review paves the way for future studies on the therapeutic potential of andrographolide, particularly as a supplementary treatment prior to chemotherapy.

Supplemental Information

Supplemental Information 1 PRISMA Checklist.

Supplemental Information 2 Rationale of study.

Supplemental Information 3 Search strategy for the role of andrographolide as a potential anticancer agent against gastric cancer cell lines.

Additional Information and Declarations

Competing Interests

Author Contributions

Data Availability

The authors declare that they have no competing interests.

Muhamad Alojid Anis Amiera conceived and designed the experiments, performed the experiments, analyzed the data, prepared figures and/or tables, authored or reviewed drafts of the article, and approved the final draft.

Tuan Noorkorina Tuan Kub conceived and designed the experiments, performed the experiments, analyzed the data, authored or reviewed drafts of the article, and approved the final draft.

Azian Harun conceived and designed the experiments, authored or reviewed drafts of the article, and approved the final draft.

Rohimah Mohamud conceived and designed the experiments, authored or reviewed drafts of the article, and approved the final draft.

Nur Rina Alissa Razian conceived and designed the experiments, analyzed the data, authored or reviewed drafts of the article, and approved the final draft.

Yasmin Khairani Muhammad Ismadi conceived and designed the experiments, analyzed the data, prepared figures and/or tables, authored or reviewed drafts of the article, and approved the final draft.

Wan Alif Syazwani Wan Alias conceived and designed the experiments, performed the experiments, analyzed the data, authored or reviewed drafts of the article, and approved the final draft.

Mohammed Hussain Abdulrazak conceived and designed the experiments, analyzed the data, authored or reviewed drafts of the article, and approved the final draft.

Ernawati Teni conceived and designed the experiments, authored or reviewed drafts of the article, and approved the final draft.

The following information was supplied regarding data availability:

This is a systematic review/meta-analysis.

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
