# Peer review of "The role of andrographolide as a potential anticancer agent against gastric cancer cell lines: a systematic review"

_PeerJ, doi:10.7717/peerj.18513_

## Round 0.1 · original submission · Major Revisions

Please respond to all the reviewers' questions thoroughly and carefully in a point-by-point manner.

Reviewer 1 ·

Basic reporting

The English used in this manuscript should be improved.
More literature and more detail introduction about gastric cancer should be added between line 49 and 54.
More specific literature should focus on the potential anti-gastric cancer about andrographolide, including clinical usage and mechanism studies, which makes audience clearly understanding why andrographolide been used in this study.

Experimental design

For a systematic review, articles covered from 2013 to 2021 is not enough. I commend to expand the scope of article search. The databases, including PubMed, Google Scholar, Web of Science, Scopus, and Science Direct should be expanded too.
The keyword “Andrographis paniculata” or “andrographolide” used to access database, actually they are different from each other.

Validity of the findings

More accurate and concise conclusions are needed.

Reviewer 2 ·

Basic reporting

The article focuses on the function of andrographolide in gastric cancer, providing current and relevant information on the issue. However, the authors should provide an explanation for what inspired them to specialize on gastric cancer. Are there any constraints or restrictions on this subject? Furthermore, the authors could highlight a broader scope of the research on andrographolide's effects in many types of cancer (i.e. cancers like breast cancer, colorectal cancer), rather than just focusing on its effect in a metastatic esophageal cancer mice model (Line 58-60). In order to strengthen the anti-cancer properties of andrographolide, it is important first look at its impact on cancer in general, before specifically focusing on its effects on gastric cancer.

In addition, the written style of the manuscript should be improved by using the correct scientific terms or phrase.
Example:
• Line 135: This review focused solely on the andrographolide in vitro activity.
Andrographolide is a test substance not an activity (it shows an effect on a certain activity i.e. tumour growth etc).
• Line 181: ..explored the in vitro anti-gastric cancer properties of AGS cells, SNU638, and SNU601..Not a suitable scientific term to be used, suggest revising as >>anti-cancer properties of andrographolide using in vitro model of gastric cancer cell lines.

It is advised the authors to send this manuscript for a scientific proofread using a professional proofreader, as to avoid from using confusing terms/statements.

The arrangement of the Results/Discussions and Conclusion was not in line with the objective of study. See the 'additional comment' section

Experimental design

1. The research question of this manuscript was not well addressed.
Line 61-62: …Despite its historical usage for gastrointestinal disorders and liver function enhancement, the evidence concerning its effectiveness and safety in the treatment of gastric cancer remains fragmented..
Does this research problem reflect the aim of this study? Seems like the study aims to focus on anti-cancer properties of andrographolide in gastric cancer. It would be essential to highlight the limitation of current study existed in in vitro effect of andrographolide in gastric cancer, as compared to the in vivo model or any other type of cancer model.

2. The method in this study was performed following the technical standard.

Validity of the findings

• Line 185-186: ..Apoptosis assays unveiled andrographolide induced apoptotic and non-apoptotic cell death , peaking at 20 µM concentration with nuclear alterations ..What is the nuclear alterations?
• Line 238-240: The authors should emphasise the justification on choosing the cell line model. Is there any study conducted to investigate the effect of andrographolide in animal model?
• Line 241-242: Irrelevant statement: The origin of stomach cancer cell lines does not accurately represent the high occurrence of cancer in that particular nation.
• Line 243: AGS cell lines. Should highlight the characteristic of this cell line as a common gastric cancer cell line used in many studies.
• Line 244: What does it mean by ‘andrographolide exhibited activity’?
• Line 245: Not sure about what is the meaning of ‘this observation is relevant’. Does the selected gastric cancer cell lines feature any different characteristics? Please explain.
• Line 244-245: ..Despite this focus, andrographolide exhibited activity against all tested gastric cell lines..What activity? Anti-proliferative activity?
• Line 248: ..All six reviewed studies reported.. but why was only one study cited?
• Line 126,187, 188, 276: rhTRAIL were mentioned in those pages but no description about its function in gastric cancer treatment. This need to be explained as to emphasise the role of andrographolide in improving the sensitivity of rhTRAIL treatment in gastric cancer.

Additional comments

The Results/Discussions and Conclusion should reflect on the objective statement in Line 65-68: The objectives encompass assessing evidence for anticancer effects, understanding underlying mechanisms, recognizing safety concerns, summarizing research findings, and pinpointing knowledge gaps.

The study aims to address the efficacy effect of andrographolide in gastric cancer, and this should be explained the results and discussions. However, it is not highlighted elsewhere in both sections.

What about discussion of andrographolide in safety concerns? What about the knowledge gap from the review analysis of the selected studies?

·

Basic reporting

The present project is a well-developed systematic review on the role of andrographolide as a potential anticancer agent against gastric cancer cell lines. The paper presents results from at least a dozen studies, of which 6 were considered relevant, after an exhaustive search in PubMed, Scopus, ScienceDirect, and other databases. This review found several studies describing the anticancer effect of andrographolide on human gastric cancer cell lines, revealing that this molecule indeed has an impact on cell proliferation and induces cytotoxicity in some articles. For this reason, andrographolide is proposed as a potential treatment for this type of cancer.

Line 100: Make sure all the letters are the same color to maintain visual consistency in the document.

Undefined acronyms:

AGS: The meaning of this acronym is not provided. Make sure to define it the first time it is mentioned in the text.
EBV (Epstein-Barr Virus): Although this acronym is used, it is not defined in the document. Include its full meaning the first time it is mentioned.
Specification of human cells: It is not specified that all the cell lines mentioned are of human origin. It would be important to clarify this to provide a clearer context for the readers.

Line 292: It is mentioned that CD markers increase, but it is not specified which ones. You should indicate specifically which CD markers are involved for greater accuracy.

Table 3: It is currently very extensive, so it is recommended to summarize its content, focusing on the most relevant or significant data for the review.

Experimental design

No comment

Validity of the findings

No comment

Additional comments

It is a good research work; the reviewed papers show the anticancer effect of the molecule andrographolide on human gastric cancer cells.

---

## Round 0.2 · accepted · Accept

The manuscript has been revised according to reviewers' comments.

Reviewer 2 ·

Basic reporting

No comment.

Experimental design

No comment

Validity of the findings

No comment

Additional comments

It is now become clear what the paper contributes to the existing literature references.